# Impact of the Sustainable Development Goals on the academic research agenda. A scientometric analysis

**Antonio Sianes[1], Alejandro Vega-Muñoz[2], Pilar Tirado-Valencia[3]\*, Antonio Ariza-Montes[4]**

1 Research Institute on Policies for Social Transformation, Universidad Loyola Andalucía, Córdoba, Spain, 2 Public Policy Observatory, Universidad Autónoma de Chile, Santiago, Chile, 3 Department of Finance and Accounting, Universidad Loyola Andalucía, Córdoba, Spain, 4 Social Matters Research Group, Universidad Loyola Andalucía, Córdoba, Spain

\* ptirado@uloyola.es

## Abstract

Today, global challenges such as poverty, inequality, and sustainability are at the core of the academic debate. This centrality has only increased since the transition from the Millennium Development Goals (MDGs) to the Sustainable Development Goals (SDGs), whose scope is to shift the world on to a path of resilience focused on promoting sustainable development. The main purpose of this paper is to develop a critical yet comprehensive scientometric analysis of the global academic production on the SDGs, from its approval in 2015 to 2020, conducted using Web of Science (WoS) database. Despite it being a relatively short period of time, scholars have published more than five thousand research papers in the matter, mainly in the fields of green and sustainable sciences. The attained results show how prolific authors and schools of knowledge are emerging, as key topics such as climate change, health and the burden diseases, or the global governance of these issues. However, deeper analyses also show how research gaps exist, persist and, in some cases, are widening. Greater understanding of this body of research is needed, to further strengthen evidence-based policies able to support the implementation of the 2030 Agenda and the achievement of the SDGs.

**Data Availability Statement:** All relevant data are within the manuscript and its Supporting Information files.

## 1. Introduction

### 1.1. From the Millennium Agenda to the 2030 Agenda and the Sustainable Development Goals (SDGs)

To track the origins of the 2030 Agenda for Sustainable Development, we must recall the Millennium Agenda, which was the first global plan focused on fighting poverty and its more extreme consequences [1]. Approved in 2000, its guiding principle was that northern countries should contribute to the development of southern states via Official Development Assistance (ODA) flows. The commitment was to reach 0.7% of donors' gross domestic product [2] to

**Funding:** The funders had no role in study design, data collection and analysis, decision to publish, or preparation of the manuscript.

**Competing interests:** The authors have declared that no competing interests exist.

reduce poverty by half by 2015. The relative failure to reach this goal and the consolidation of a discourse of segregation between northern and southern countries [3] opened the door to strong criticism of the Millennium Agenda. Therefore, as 2015 approached, there were widespread calls for a profound reformulation of the system [4].

The world in 2015 was very different from that in the early 2000s. Globalization had reached every corner of the world, generating development convergence between countries but increasing inequalities within countries [5, 6]. Increasing interest in the environmental crisis and other global challenges, such as the relocation of work and migration flows, consolidated a new approach to development and the need of a more encompassed agenda [7]. This new agenda was conceived after an integrating process that involved representatives from governments, cooperation agencies, nongovernmental organisations, global business, and academia. The willingness of the 2030 Agenda to 'leave no one behind' relies on this unprecedented global commitment by the international community [8].

As a result of this process, in 2015, the United Nations General Assembly formally adopted the document "Transforming our World: the 2030 Agenda for Sustainable Development" [9], later known as the 2030 Agenda. This new global agenda is an all-comprising strategy that seeks to inform and orient public policies and private interventions in an extensive range of fields, from climate change to smart cities and from labour markets to birth mortality, among many others.

The declared scope of the Agenda is to shift the world on to a path of resilience focused on promoting sustainable development. To do so, the 2030 Agenda operates under the guidance of five principles, formally known as the '5 Ps': people, planet, prosperity, peace, and partnerships [10]. With these pivotal concepts in mind, the Agenda has established a total of 17 Sustainable Development Goals (SDGs) and 169 specific targets to be pursued in a 15-year period, which reflects the scale and profound ambition of this new Agenda.

The SDGs do not only address what rich countries should do for the poor but rather what all countries should do together for the global well-being of this and future generations [4]. Thus, the SDGs cover a much broader range of issues than their predecessors, the Millennium Development Goals [11], and are intended to be universal on the guidance towards a new paradigm of sustainable development that the international community has been demanding since the 1992 Earth Summit [7, 12, 13].

Despite this potential, some criticise their vagueness, weakness, and unambitious character. Fukuda-Parr [14], see weaknesses on the simplicity of the SDGs, which can lead to a very narrow conception that reduces the integral concept of development. The issue of measurement is also problematic; for some researchers, the quantification of objectives not only reduces their complexity, but leads to them being carried out without considering the interdependencies between the objectives [12, 13]. Other authors have identified difficulties associated with specifying some of the less visible, intangible aspects of their qualitative nature such as inclusive development and green growth [14, 15]. Finally, Stafford-Smith et al. [16] state that their successful implementation also requires paying greater attention to the links across sectors, across societal actors and between and among low-, medium-, and high-income countries.

Despite these criticisms, the SDGs have undoubtedly become the framework for what the Brundtland report defined as our common future. Unlike conventional development agendas that focus on a restricted set of dimensions, the SDGs provide a holistic and multidimensional view of development [17]. In this line, Le Blanc [12] concludes that the SDGs constitute a system with a global perspective; because they consider the synergies and trade-offs between the different issues involved in sustainable development, and favour comprehensive thinking and policies.

## 1.2. Towards a categorization of the SDGs

There is an underlying lack of unanimity in the interpretation of the SDGs, which has given rise to alternative approaches that allow categorizing the issues involved in their achievement without losing sight of the integral vision of sustainable development [15, 18–23]. However, such categorization of the SDGs makes it possible to approach them in a more holistic and integrated way, focusing on the issues that underlie sustainable development and on trying to elucidate their connections.

Among the many systematization proposals, and following the contributions of Hajer et al. [19], four connected perspectives can strengthen the universal relevance of the SDGs: a) 'planetary boundaries' that emphasize the urgency of addressing environmental concerns and calling on governments to take responsibility for global public goods; b) 'The safe and just operating space' to highlight the interconnectedness of social and environmental issues and their consequences for the redistribution of wealth and human well-being; c) 'The energetic society' that avoids the plundering of energy resources; and d) 'green competition' to stimulate innovation and new business practices that limit the consumption of resources.

Planetary boundaries demand international policies that coordinate efforts to avoid overexploitation of the planet [24]. Issues such as land degradation, deforestation, biodiversity loss and natural resource overexploitation exacerbate poverty and deepen inequalities [21, 25–27]. These problems are further compounded by the increasing impacts of climate change with clear ramifications for natural systems and societies around the globe [21, 28].

A safe and just operating space implies social inclusivity that ensures equity principles for sharing opportunities for development [15, 29]. Furthermore, it requires providing equitable access to effective and high-quality preventive and curative care that reduces global health inequalities [30, 31] and promotes human well-being. Studies such as that of Kruk et al. [32] analyse the reforms needed in health systems to reduce mortality and the systemic changes necessary for high-quality care.

An energetic society demands global, regional and local production and consumption patterns as demands for energy and natural resources continue to increase, providing challenges and opportunities for poverty reduction, economic development, sustainability and social cohesion [21].

Finally, green competition establishes limits to the consumption of resources, engaging both consumers and companies [22] and redefining the relationship between firms and their suppliers in the supply chain [33]. These limits must also be introduced into life in cities, fostering a new urban agenda [34, 35]. Poor access to opportunities and services offered by urban centres (a function of distance, transport infrastructure and spatial distribution) is a major barrier to improved livelihoods and overall development [36].

The diversification of development issues has opened the door to a wide range of new realities that must be studied under the guiding principles of the SDGs, which involve scholars from all disciplines. As Saric et al. [37] claimed, a shift in academic research is needed to contribute to the achievement of the 2030 Agenda. The identification of critical pathways to success based on sound research is needed to inform a whole new set of policies and interventions aimed at rendering the SDGs both possible and feasible [38].

## 1.3. The relevance and impact of the SDGs on academic research

In the barely five years since their approval, the SDGs have proven the ability to mobilize the scientific community and offer an opportunity for researchers to bring interdisciplinary knowledge to facilitate the successful implementation of the 2030 Agenda [21]. The holistic vision of development considered in the SDGs has impacted very diverse fields of knowledge,

such as land degradation processes [25, 26], health [39], energy [40] and tourism [41], as well as a priori further disciplines such as earth observation [42] and neurosurgery [43]. However, more importantly, the inevitable interdependencies, conflicts and linkages between the different SDGs have also emerged in the analyses, highlighting ideas such as the need for systemic thinking that considers the spatial and temporal connectivity of the SDGs, which calls for multidisciplinary knowledge. According to Le Blanc [12], the identification of the systemic links between the objectives can be a valuable undertaking for the scientific community in the coming years and sustainable development.

Following this line, several scientific studies have tried to model the relationships between the SDGs in an attempt to clarify the synergies between the objectives, demonstrating their holistic nature [12, 17, 20, 44, 45]. This knowledge of interdependencies can bring out difficulties and risks, or conversely the drivers, in the implementation of the SDGs, which will facilitate their achievement [22]. In addition, it will allow proposing more transformative strategies to implement the SDG agenda, since it favours an overall vision that is opposed to the false illusion that global problems can be approached in isolation [19].

The lack of prioritisation of the SDGs has been one of the issues raised regarding their weakness, which should also be addressed by academics. For example, Gupta and Vegelin [15] analyse the dangers of inclusive development prioritising economic issues, relegating social or ecological inclusivity to the background, or the relational aspects of inclusivity that guarantee the existence of laws, policies and global rules that favour equal opportunities. Holden et al. [46] suggest that this prioritisation should be established according to three moral criteria: the satisfaction of human needs, social equity and respect for environmental limits. These principles must be based on ethical values that, according to Burford et al. [47], constitute the missing pillar of sustainability. In this way, the ethical imperatives of the SDGs and the values implicit in the discourses on sustainable development open up new possibilities for transdisciplinary research in the social sciences [46, 47].

Research on SDG indicators has also been relevant in the academic world, as they offer an opportunity to replace conventional progress metrics such as gross domestic product (GDP) with other metrics more consistent with the current paradigm of development and social welfare that takes into account such aspects as gender equality, urban resilience and governance [20, 48].

The study of the role of certain development agents, including companies, universities or supranational organisations, also opens up new areas of investigation for researchers. Some studies have shown the enthusiastic acceptance of the SDGs by companies [22, 49]. For Bebbington and Unerman [50], the study of the role of organisations in achieving the SDGs should be centred around three issues: challenging definitions of entity boundaries to understand their full impacts, introducing new conceptual frameworks for analysis of the context within which organisations operate and re-examining the conceptual basis of justice, responsibility and accountability. On the other hand, the academic community has recognized that knowledge and education are two basic pillars for the transition towards sustainable development, so it may also be relevant to study the responsibility of higher education in achieving the SDGs [47, 50]. Institutional sustainability and governance processes are issues that should be addressed in greater depth through research [47].

Finally, some authors have highlighted the role of information technologies (ICT) in achieving the SDGs [23] and their role in addressing inequality or vulnerability to processes such as financial exclusion [51], which opens up new avenues for research.

Despite this huge impact of the SDGs on academic research, to the best of our knowledge, an overall analysis of such an impact to understand its profoundness and capillarity is missing in the literature. To date, reviews have focused on the implementation of specific SDGs

[52–61], on specific topics and collectives [62–70], on traditional fields of knowledge, now reconsidered in light of the SDGs [71–73] and on contributions from specific regions or countries [74, 75]. By relying on scientometric techniques and data mining analyses, this paper collects and analyses the more than 5,000 papers published on the SDGs to pursue this challenging goal and fill this knowledge gap.

This article aims to provide a critical review of the scientific research on SDGs, a concept that has emerged based on multiple streams of thinking and has begun to be consolidated as of 2015. As such, global references on this topic are identified and highlighted to manage preexisting knowledge to understand relationships among researchers and with SDG dimensions to enhance the presently dispersed understanding of this subject and its areas of further development. A scientometric meta-analysis of publications on SDGs is conducted to achieve this objective. Mainstream journals from the Web of Science (WoS) are used to identify current topics, the most involved journals, the most prolific authors, and the thematic areas around which the current academic SDG debate revolves.

Once Section 1 has revised on the related literature to accomplish the main objective, Section 2 presents the research methodology. Section 3 presents the main results obtained, and Section 4 critically discusses these results. The conclusion and the main limitations of the study are presented in Section 5.

## 2. Materials and methods

In methodological terms, this research applies scientometrics as a meta-analytical means to study the evolution of documented scientific knowledge on the Sustainable Development Goals [76–81], taking as a secondary source of information academic contributions (i.e. articles, reviews, editorials, etc.) indexed in the Web of Science (WoS). To ensure that only peer-reviewed contributions authored by individual researchers are retrieved and that such publications have a worldwide prestige assessment, all of them should be published on journals indexed in the Journal Citation Report (JCR), either as part of the Sciences Citation Index Expanded or the Social Sciences Citation Index [82–84].

Following the recommendations of previous studies [85], it was decided to apply the next search vector from 2015 to 2020 to achieve the research objectives TS = (Sustainable NEAR/0 Development NEAR/0 Goals), which allows the extraction of data with 67 fields for each article registered in WoS.

As the first step, to give meaning to subsequent analyses, we tested the presence of exponential growth in the production of documented knowledge that allows a continuous renewal of knowledge [76, 86].

As a second action, given the recent nature of the subject studied, it is of interest to map the playing field [87] using VOSviewer software version 1.6.16 [88], to know which topics are most addressed in the matter of SDGs. This analysis seeks an approach, both through the concentration of Keyword Plus® [89] and by analysing the references used as input in the production of knowledge, which can be treated as cocitations, coupling-citations and cross-citations [90], using the h-index, in citation terms, as discriminant criteria in the selection of articles [91–93]. This methodology will allow us to establish production, impact and relationship metrics [80, 85, 87, 94, 95].

Finally, it is of interest to explore the possible concentrations that may arise. Using Lotka's Law, we estimated the possible prolific authors and their areas of work in SDGs, and using Bradford's Law, we conducted a search of a possible adjustment to a geometric series of the concentration zones of journals and therefore a potential nucleus where a profuse discussion on SDGs is taking place [96–100].

**Table 1. Top ten JCR-WoS categories publishing literature on SDGs.**

| Ranking | JCR-WoS Category | % Articles | Index |
|---|---|---|---|
| 1 | Environmental Sciences | 32.49% | SCI-E |
| 2 | Green Sustainable Science Technology | 21.41% | SCI-E |
| 3 | Environmental Studies | 19.21% | SCI-E |
| 4 | Public Environmental Occupational Health | 14.53% | SCI-E |
| 5 | Engineering Environmental | 4.69% | SCI-E |
| 6 | Development Studies | 4.43% | SSCI |
| 7 | Water Resources | 4.40% | SCI-E |
| 8 | Economics | 4.00% | SSCI |
| 9 | Multidisciplinary Sciences | 3.63% | SCI-E |
| 10 | Medicine General Internal | 3.47% | SCI-E |

# 3. Results

## 3.1. Configuration of the academic production on SDGs

The results present a total of 5,281 articles for a period of six years (2015–2020) in 1,135 journals, with over 60% of these documents published in the last two years. The total of articles is distributed among authors affiliated with 7,418 organisations from 181 countries/regions, giving thematic coverage to 183 categories of the Journal Citation Report-Web of Science (JCR-WoS). Table 1 shows the distribution among the top ten JCR-WoS categories, highlighting the prevalence of journals indexed in green and environmental sciences and, thus, in the Science Index-Expanded.

## 3.2. Existence of research critical mass

Fig 1 shows the regression model for the period 2015–2020, the last year with complete records consolidated in the Web of Science. The results obtained show significant growth in the

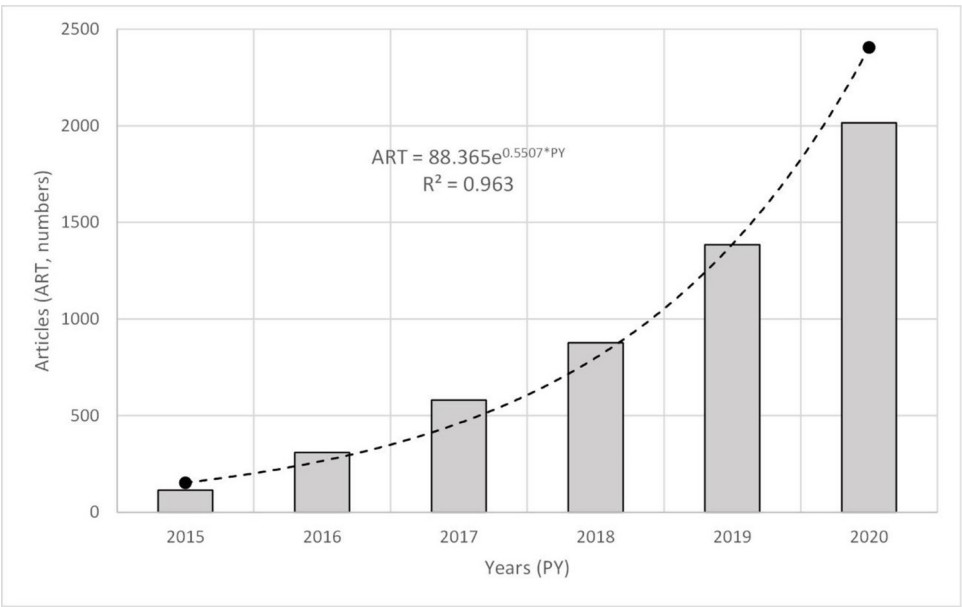

$$ART = 88.365e^{0.5507*PY}$$
$$R^2 = 0.963$$

**Fig 1. Academic production annual growth, data source WoS, 2020.**

**Table 2. Contemporary prolific authors, 2019–2020.**

| Authors | Articles | Affiliation | Frequented journals (2019–2020) | SDG Dimension |
|---|---|---|---|---|
| Abhilash, PC | 12 | Banaras Hindu Univ | Agronomy-Basel, Bioresour. Technol., Ecol. Indic., Environ. Dev., J. Clean Prod., Land (2), Land Degrad. Dev., Land Use Pol., Restor. Ecol., Sci. Total Environ., Sustain. Sci. | Environment |
| Murray, CJL | 12 | Univ. of Washington | JAMA Oncol., JAMA Pediatr. Lancet (6), Lancet Glob. Health, Lancet Public Health (2), Nature. | Health |
| Leal-Filho, W | 11 | Hamburg Univ of Applied Sciences, Manchester Metropolitan Univ | Int. J. Sustain. Dev. World Ecol. (2), Int. J. Sustain. High. Educ. (2), Int. J. Sustain. High. Educ. (2), J. Clean Prod. (5), Sustainability (2). | Environment |
| Yaya, S | 11 | Univ Ottawa, Univ Oxford, Univ Parakou | Arch. Public Health, Biomed Res. Int., BMC Infect. Dis., BMC Pregnancy Childbirth, BMC Public Health (2), J. Glob. Health, Lancet (2), Reprod. Health (2). | Health |
| Bhutta, ZA | 10 | Hosp Sick Children. AGA Khan Univ. | Am. J. Clin. Nutr., BMC Public Health, BMJ Glob. Health (2), Clin. Infect. Dis., JAMA Netw. Open, JAMA Pediatr., Lancet, Lancet Glob. Health, Nature. | Health |
| Kalin, RM | 10 | Univ Strathclyde | Appl. Sci.-Basel (2), Environ. Sci.-Wat. Res. Technol., J. Hydrol.-Reg. Stud., Sci. Total Environ. (2), Water (4). | Environment |
| Alola, AA | 9 | Istanbul Gelisim Univ, South Ural State Univ, Eastern Mediterranean Univ | Bus. Strateg. Environ., Energy Policy, Environ. Sci. Pollut. Res., Sci. Total Environ. (5), Sustain. Dev. | Environment + Economics |
| Hay, SI | 9 | Univ. of Washington | JAMA Oncol., JAMA Pediatr., Lancet. (4), Lancet Glob. Health, Nature (2). | Health |

number of studies on SDGs, with an $R^2$ adjustment greater than 96%. The exponential nature of the model shows that a 'critical mass' is consolidating around the research on this topic, as proposed by the Law of Exponential Growth of Science over Time [76], which in some way gives meaning to this research and to obtaining derived results.

## 3.3. Establishment of concentrations

In accordance with Lotka's Law, 22,336 authors were identified of the 5,281 articles under study. From this author set, 136 ($\approx$sqrt (22,336)) are considered prolific authors with a contribution to nine or more works. However, a second restriction, even more demanding, is to identify those prolific authors who are also prolific in contemporary terms. Although SDG studies are recent, the growth production rates are extremely high. As previously shown, for the period 2015–2020, 64% of the publications are concentrated between 2019–2020. Based on this second restriction, for 3,400 articles of the 5,281 articles published in 2019 and 2020, and a total of 15,120 authors, only eight prolific authors manage to sustain a publication number that equals or exceeds nine articles. These authors are listed and characterized in Table 2.

The analysis shown in Table 2 highlights the University of Washington's participation in health issues with Murray and Hay (coauthors of eight articles in the period 2019–2020), who are also important in the area of health for the prolific authors Yaya and Bhutta. The environmental SDGs mark a strong presence with Abhilash, Leal-Filho and Kalin. The affiliation of Abhilsash (Banaras Hindu University) is novel, as it is not part of the classic world core in knowledge production that is largely concentrated in the United States and Europe. It is worth noting that other prolific authors belong to nonmainstream knowledge production world areas, such as Russia or Pakistan. Professor Alola also deserves mention; not only is he the only contemporary prolific author producing in the area of economics, but he is also producing knowledge in Turkey.

In the same way, at the journal level, the potential establishment of concentration areas and determination of a deep discussion nucleus are analysed using Bradford's law.

With a percentage error of 0.6%, between the total journal number and the total journal number estimated by the Bradford series, the database shows a core of 18 journals (2%) where one in three articles published are concentrated (see Table 3).

**Table 3. Bradford zoning.**

| Zone | # Articles (%) | | Journals (%) | | Bradford multipliers | Bradford Series |
|---|---|---|---|---|---|---|
| Nucleus | 1,742 | (33%) | 18 | (2%) | | 18 |
| 1 | 1,744 | (33%) | 128 | (12%) | 7.1 | 134 |
| 2 | 1,795 | (34%) | 989 | (86%) | 7.7 | 991 |
| Total/*Mean* | 5,281 | (100%) | 1,135 | | 7.4 | 1,142 |

Regarding the number of contributions by journal, Sustainability has the largest number of studies on SDGs, in which 689 (13%) of the 5,281 articles studied are concentrated. The Journal of Cleaner Production, indexed to WoS categories related to Environmental SDGs, is the second most prominent journal, with 2.7% participation of the articles (147). Both journals are followed by the multidisciplinary journal Plos One, with 2.2% of the total dataset. In terms of impact factor, the 60 points of the health journal The Lancet are superlative in the whole, which in the other cases ranges between 2.000 and 7.246. As shown in Table 4, we have developed a "Prominence ranking" by weighting article production by impact factor. This metric shows The Lancet, with only 40 articles on SDGs, as the most relevant journal, followed by Sustainability, which becomes relevant due to the high number of publications (689) despite an impact factor of 2.576. These journals are followed by the Journal of Cleaner Production with 147 articles and an impact factor of 7.246.

**Table 4. Bradford nucleus journals.**

| Source | # Articles (ART) | WoS Categories | IF 2019 | Prominence Ranking |
|---|---|---|---|---|
| *Lancet* | 40 | Medicine. General & Internal | 60.390 | 1 |
| *Sustainability* | 689 | Green & Sustainable Science & Technology; Environmental Sciences; Environmental Studies | 2.576 | 2 |
| *Journal of Cleaner Production* | 147 | Green & Sustainable Science & Technology; Engineering. Environmental; Environmental Sciences | 7.246 | 3 |
| *Science of The Total Environment* | 85 | Environmental Sciences | 6.551 | 4 |
| *BMJ Global Health* | 74 | Public. Environmental & Occupational Health | 4.280 | 5 |
| *Plos One* | 114 | Multidisciplinary Sciences | 2.740 | 6 |
| *Sustainability Science* | 56 | Green & Sustainable Science & Technology; Environmental Sciences | 5.301 | 7 |
| *World Development* | 68 | Development Studies; Economics | 3.869 | 8 |
| *Sustainable Development* | 56 | Development Studies; Green & Sustainable Science & Technology; Regional & Urban Planning | 4.082 | 9 |
| *International Journal of Environmental Research and Public Health* | 72 | Environmental Sciences; Public. Environmental & Occupational Health | 2.849 | 10 |
| *Environmental Science & Policy* | 41 | Environmental Sciences | 4.767 | 11 |
| *Journal of Sustainable Tourism* | 38 | Green & Sustainable Science & Technology; Hospitality, Leisure, Sport & Tourism | 3.986 | 12 |
| *Marine Policy* | 44 | Environmental Studies; International Relations | 3.228 | 13 |
| *Water* | 48 | Water Resources | 2.544 | 14 |
| *BMC Public Health* | 48 | Public. Environmental & Occupational Health | 2.521 | 15 |
| *International Journal of Sustainable Development and World Ecology* | 39 | Green & Sustainable Science & Technology; Ecology | 2.772 | 16 |
| *Global Health Action* | 46 | Public. Environmental & Occupational Health | 2.162 | 17 |
| *International Journal of Sustainability in Higher Education* | 37 | Green & Sustainable Science & Technology; Education & Educational Research | 2.000 | 18 |
| *Total* | 1,742 | Mean | 6.881 | |

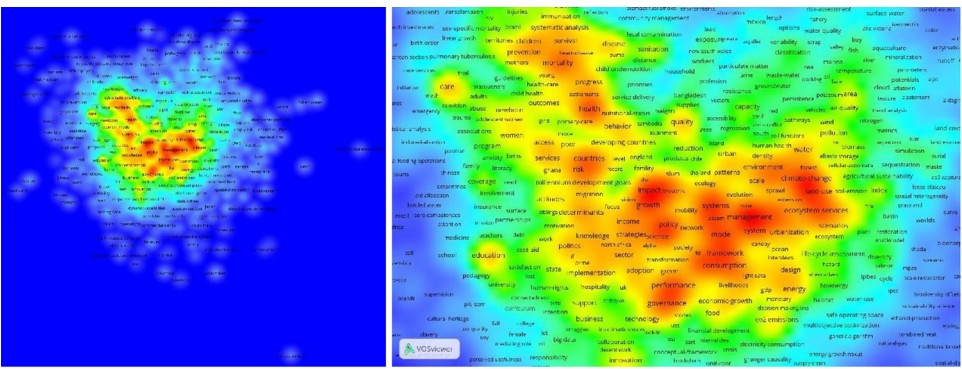

**Fig 2.** a) Keywords Plus® heatmap and b) heat map zoom to highlight the highest concentration words, data source WoS, 2020.

## 3.4. Thematic coverage

Concerning the thematic coverage, Fig 2A and 2B show a diversity of 7,003 Keyword Plus® (KWP), consistently connected to a total of 7,141 KWP assigned by Clarivate as metadata to the set of 5,281 articles studied, which presents a strong concentration in a small number of terms (red colour in the heat map generated with VOSviewer version 1.6.16).

Based on this result, a concentration sphere with 85 KWP (= sqrt (7,141)) is established according to Zipf's Law, which is presented in 50 or more articles out of the total of 5,281. Moreover, a central concentration sphere of 9 KWPs (= sqrt (85)) can be found, with keywords present in a range of 178 to 346 articles out of a total of 5,281. These nine pivotal keywords are all connected in terms of co-occurrence (associated by Clarivate two or more to the same article) and within papers with an average number of citations in WoS that vary from 9.27 to 16.69, as shown in Table 5. The nine most prominent key words in relation to the study of the SDGs are health, climate change, management, impact, challenges, governance, systems, policy and framework. These terms already suggest some of the themes around which the debate and research in this area revolves.

The prominence of these keywords is obtained by combining the level of occurrence and average citations (see Table 5): on the one hand, the occurrence or number of articles with which the KWP is associated (e.g., Management, 346) and, on the other hand, the average citations presented by the articles associated with these words (e.g., Framework. 9.27). The final

**Table 5. Outstanding KWP in the database.**

| KWP | Occurrences | Average citations | Prominence | Ranking |
|---|---|---|---|---|
| Health | 330 | 16.69 | 190% | 1 |
| Climate-change | 255 | 16.16 | 142% | 2 |
| Management | 346 | 10.32 | 123% | 3 |
| Impact | 292 | 11.91 | 120% | 4 |
| Challenges | 187 | 12.25 | 79% | 5 |
| Governance | 207 | 10.95 | 78% | 6 |
| Systems | 195 | 9.76 | 66% | 7 |
| Policy | 178 | 10.35 | 64% | 8 |
| Framework | 190 | 9.27 | 61% | 9 |
| Mean | 246 | 11.96 | 100% | |
| Standard deviation | 66 | 2.70 | | |

score (prominence) mixes both concepts, given the product of the occurrences and the average citations of each KWP in proportion to the mean values (e.g., (330 * 16.69)/(246 * 11.96) = 1.9).

## 3.5. Relations within the academic contributions

The coupling-citation analysis using VOSviewer identifies the 5,281 articles under study, of which only those found in the h-index as a whole have been considered (the h-index in the database is 81, as there are 81 articles cited 81 or more times). The bibliographic coupling analysis found consistent connections in only 73 of these articles, gathered in seven clusters. Such clusters and unconnected articles are represented in Fig 3.

In simple terms, discrimination belonging to one cluster or another depends on the total link number that an article has with the other 80 articles based on the use of the common references. Table 6 specifies the articles belonging to the same publication cluster in relation to Fig 3.

Bibliographic coupling analysis can also be used to link the seven clusters that use common references with the field document title (TI), publication name (SO), Keyword Plus-KWP (ID), and research areas (SC). This allows the identification of the main topics of each cluster. As shown in Table 7, cluster 1 (red) concerns environmental and public affairs; cluster 2 (green), health; cluster 3 (blue), economics; cluster 4 (yellow), health–the burden of disease; cluster 5 (violet), economics–Kuznets curve; cluster 6 (light blue), energy; and cluster 7 (orange), soil—land.

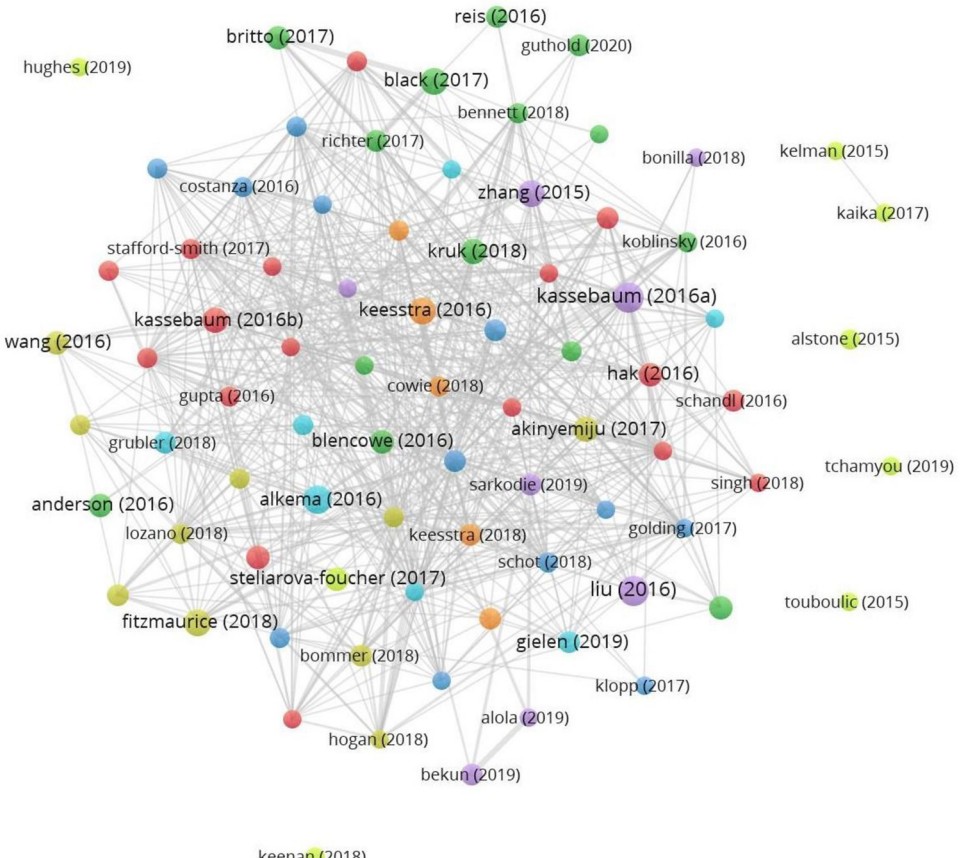

**Fig 3. Coupling-citation graph.** Data source WoS. 2020.

**Table 6. Articles according to coupling-citation clusters.**

| Cluster | # items | Articles |
|---------|---------|----------|
| 1 (Red) | 17 | Bebbington (2018); Gao (2017); Gupta (2016); Hajer (2015); Hak (2016); Hickel (2020); Holden (2017); Kassebaum (2016b); Le Blanc (2015); Obersteiner (2016); Pradhan (2017); Schandl (2016); Scheyvens (2016); Singh (2018); Stafford-Smith (2017); Wood (2018); Wu (2018). |
| 2 (Green) | 14 | Anderson (2016); Bennett (2018); Black (2017); Blencowe (2016); Britto (2017); Guthold (2020); Hanson (2015); Koblinsky (2016); Kruk (2018); Norheim (2015); Reis (2016); Richter (2017); Shiels (2017); You (2015). |
| 3 (Blue) | 12 | Costanza (2016); Golding (2017); Klopp (2017); Kubiszewski (2017); Lim (2016); Parnell (2016); Rasul (2016); Schot (2018); Schroeder (2019); Stenberg (2017); Thilsted (2016); Weiss (2018). |
| 4 (Yellow) | 10 | Akinyemiju (2017); Bommer (2018); Fitzmaurice (2018); Fitzmaurice (2019); Fullman (2017); Hogan (2018); Lozano (2018); Luyckx (2018); Nayagam (2016); Wang (2016). |
| 5 (Violet) | 8 | Alola (2019); Bekun (2019); Bonilla (2018); Kassebaum (2016a); Liu (2016); Sarkodie (2019); Shahbaz (2019); Zhang (2015). |
| 6 (Light Blue) | 7 | Ali (2017); Alkema (2016); Chaudhary (2018); Gielen (2019); Grubler (2018); McCollum (2018); Sachs (2019). |
| 7 (Orange) | 5 | Bryan (2018); Cowie (2018); Keesstra (2016); Keesstra (2018); Xu (2017). |

## 3.6. Outstanding contributions in the field

The cocitation analysis identified a total of 232,081 references cited by the 5,281 articles under study. It suggests taking as references to review those that present 44 or more occurrences in the database (232,081/5,281). This method results in 34 articles that have been used as main inputs for the scientific production under analysis, cited between 44 and 504 times. A result worth highlighting is that one in three of these documents corresponds to reports from international organisations, such as the United Nations (UN), United Nations Educational, Scientific and Cultural Organization (UNESCO), United Nations International Children's Emergency Fund (UNICEF), United Nations Fund for Population Activities (UNFPA), World Bank Group (WB) or World Health Organization (WHO). However, it is also possible to identify 21 peer-reviewed scientific contributions. These papers are identified in detail in Table 8.

The cocitation analysis yields the degree of relationship of these 21 most cited research articles. It is how such references have been used simultaneously in the same article. Fig 4 displays this information (to help readers, it has also been included in Table 8, centrality in 21 column).

According to the relationship level in the most cited article's selection, the graph (Fig 3) has been clustered in three colours: cluster 1 in red colour groups the highest articles proportion (9) published between 2013 and 2017 in 7 journals. These journals present an impact factor (IF) quite heterogeneous, with values ranging from 2.576 (Sustainability) to 60.39 (Lancet) and indexed in one or more of the following WoS categories: Environmental Sciences (4 journals), Green & Sustainable Science & Technology (4), Environmental Studies (2), Development Studies (1), Medicine, General & Internal (1), Multidisciplinary Sciences (1) and Regional & Urban Planning (1). Three of these articles are cited 130–150 times in the 5,281-article dataset and, at the same time, show a connection centrality of 95–100% with the other 20 articles in the graph, implying a high level of cocitation. The other two clusters group six articles each. The articles of cluster 2 (green colour) are included in a widespread WoS category set: Environmental Sciences (3 journals), Geosciences, Multidisciplinary (2), Ecology (1), Economics (1), Energy & Fuels (1), Environmental Studies (1), Green & Sustainable Science & Technology (1), Materials Science, Multidisciplinary (1), Meteorology & Atmospheric Sciences (1) and Multidisciplinary Sciences (1). The research of Nilsson [101] was used as a reference in 176 of the 5,281 articles under study, showing a centrality of 100%. This great connection level is also

**Table 7. Articles according to coupling-citation clusters.**

| Cluster | # items | Identified Patterns (Fields: TI, SO, ID, SC) | Key topics |
|---|---|---|---|
| 1 (Red) | 17 | TI: no identified patterns | Environmental; Public Affairs. |
| | | SO: Sustainable Development (3). | |
| | | ID: Climate-Change (4); Governance (4); Management (3); Adaptation (2); Biodiversity (2); Ecosystem Services (2); Energy (2). | |
| | | SC: Environmental Sciences & Ecology (8); Science & Technology—Other Topics (8); Business & Economics (3); Development Studies (3); Public Administration (3); Government & Law (2); International Relations (2). | |
| 2 (Green) | 14 | TI: Trends (4); Early childhood development (3). | Health. |
| | | SO: Lancet (10); Lancet Global Health (2). | |
| | | ID: no identified patterns. | |
| | | SC: General & Internal Medicine (10); Public, Environmental & Occupational Health (2). | |
| 3 (Blue) | 12 | TI: Economics (Circular economy, R&D, inequality, financing, future value, managing, defining agenda) and quantitative (indicators, measuring, modelling, mortality) terms. | Economics. |
| | | SO: Lancet (2). | |
| | | ID: no identified patterns. | |
| | | SC: Business & Economics (4); Environmental Sciences & Ecology (4); General & Internal Medicine (2); Science & Technology—Other Topics (2). | |
| 4 (Yellow) | 10 | TI: Global (9); Burden (8). | Health–Burden of Disease. |
| | | SO: Jama Oncology (3); Lancet (2). | |
| | | ID: no identified patterns. | |
| | | SC: Oncology (3); General & Internal Medicine (2); Infectious Diseases (2); Public, Environmental & Occupational Health (2). | |
| 5 (Violet) | 8 | TI: Economics (Economic growth, Economic development, Foreign direct investment, Industry 4.0, Managing) terms. | Economics–Kuznets curve. |
| | | SO: Science of The Total Environment (3); Lancet (2). | |
| | | ID: Kuznets curve (5); China (3); Australia (2); $CO_2$ Emissions (3). | |
| | | SC: Environmental Sciences & Ecology (5); Science & Technology—Other Topics (3); General & Internal Medicine (2). | |
| 6 (Light Blue) | 7 | TI: Energy (3). | Energy. |
| | | SO: Nature Energy (2). | |
| | | ID: Greenhouse-Gas Emissions (2); Wind (2); Other sources energy (Carbon, Hydrogen). | |
| | | SC: Energy & Fuels (3); Science & Technology—Other Topics (3); Materials Science (2). | |
| 7 (Orange) | 5 | TI: Land (2); Land Degradation Neutrality (2); Soil (2). | Soil–Land. |
| | | SO: no identified patterns | |
| | | ID: Ecosystem Services (3); Erosion (3). | |
| | | SC: Environmental Sciences & Ecology (2); Science & Technology—Other Topics (2). | |

featured in another less cited article [17] published in Earth's Future. Finally, cluster 3 (blue) highlights six articles concentrated in three highly cited journals in the WoS categories: Medicine, General & Internal (Lancet) and Multidisciplinary Sciences (Nature and Science), whose IFs range from 41.9 to 60.4. In general, they are articles less connected (cocited) to the set of 21, with centralities of 30–90%. Two of these articles were referenced 140 times or more, although one was published in 2009. Thus, cluster 3 concentrates the references mainly in journals on environmental issues with scientific-technological orientation, as well as classic and high-impact WoS journals (The Lancet, Nature and Science). It is worth noting that some of these top journals may not be listed in Table 4 as they are not included in the Bradford's nucleus, due to their comparatively low number of contributions published.

Finally, continuing with the thematic study, a cross-citation analysis was developed. Considering only the 81 articles that are part of the h-index of the total set of 5,821 articles under study, the citations that are presented among this elite article set are explored using

**Table 8. Articles cocited as references in the set of articles studied.**

| First author | Year | Journal | IF 2019 | Type of contribution | Citations in 5,281 | Centrality in 21 | Cluster |
|---|---|---|---|---|---|---|---|
| Nilsson M | 2016 | Nature | 42.779 | Editorial | 176 | 100% | 2 (green) |
| Griggs D | 2013 | Nature | 42.779 | Editorial | 152 | 100% | 1 (red) |
| Steffen W | 2015 | Science | 41.846 | Article | 145 | 90% | 3 (blue) |
| Rockstrom J | 2009 | Nature | 42.779 | Article | 140 | 85% | 3 (blue) |
| Sachs JD | 2012 | Lancet | 60.390 | Editorial | 135 | 95% | 1 (red) |
| Le Blanc D | 2015 | Sustainable Development | 4.082 | Article | 133 | 95% | 1 (red) |
| Hak T | 2016 | Ecological Indicators | 4.229 | Article | 83 | 90% | 1 (red) |
| Pradhan P | 2017 | Earth's Future | 6.141 | Article | 82 | 100% | 2 (green) |
| Stafford-Smith M | 2017 | Sustainability Science | 5.301 | Article | 66 | 90% | 1 (red) |
| Nerini FF | 2018 | Nature Energy | 46.495 | Review | 55 | 80% | 2 (green) |
| Costanza R | 2016 | Ecological Economics | 4.482 | Article | 52 | 85% | 2 (green) |
| Foley JA | 2011 | Nature | 42.779 | Article | 52 | 80% | 3 (blue) |
| Scheyvens R | 2016 | Sustainable Development | 4.082 | Article | 52 | 60% | 1 (red) |
| Black RE | 2013 | Lancet | 60.390 | Article | 49 | 30% | 3 (blue) |
| Lim SS | 2016 | Lancet | 60.390 | Article | 49 | 65% | 3 (blue) |
| Hajer M | 2015 | Sustainability | 2.576 | Article | 48 | 80% | 1 (red) |
| Lu YL | 2015 | Nature | 42.779 | Editorial | 47 | 85% | 1 (red) |
| Biermann F | 2017 | Current Opinion in Env. Sustainability | 5.658 | Review | 46 | 80% | 1 (red) |
| Weitz N | 2018 | Sustainability Science | 5.301 | Article | 46 | 80% | 2 (green) |
| Schmidt-Traub G | 2017 | Nature Geoscience | 13.566 | Article | 45 | 90% | 2 (green) |
| Godfray HCJ | 2010 | Science | 41.846 | Review | 45 | 50% | 3 (blue) |

VosViewer. The cross-citation analysis detects existing relationships between 37 of these 81 articles. Once the directionality of the citations has been analysed, a directed temporal graph is generated using Pajek 64 version 5.09, which is presented in Fig 5.

Fig 5 shows how these 37 highly cited articles are related to each other (the number after the name is the publication year), considering that some of these articles are cited as references

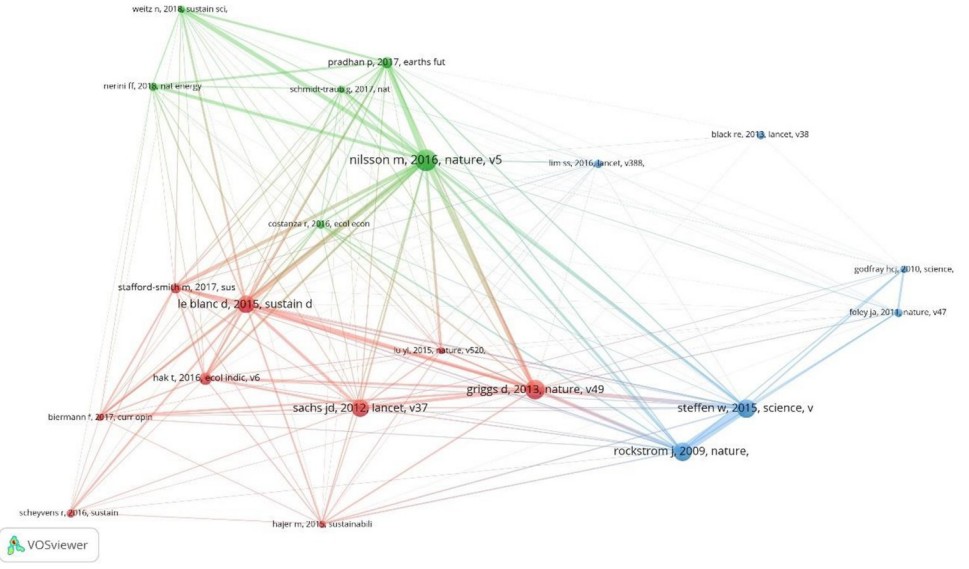

**Fig 4. Cocitation graph, data source WoS.** 2020.

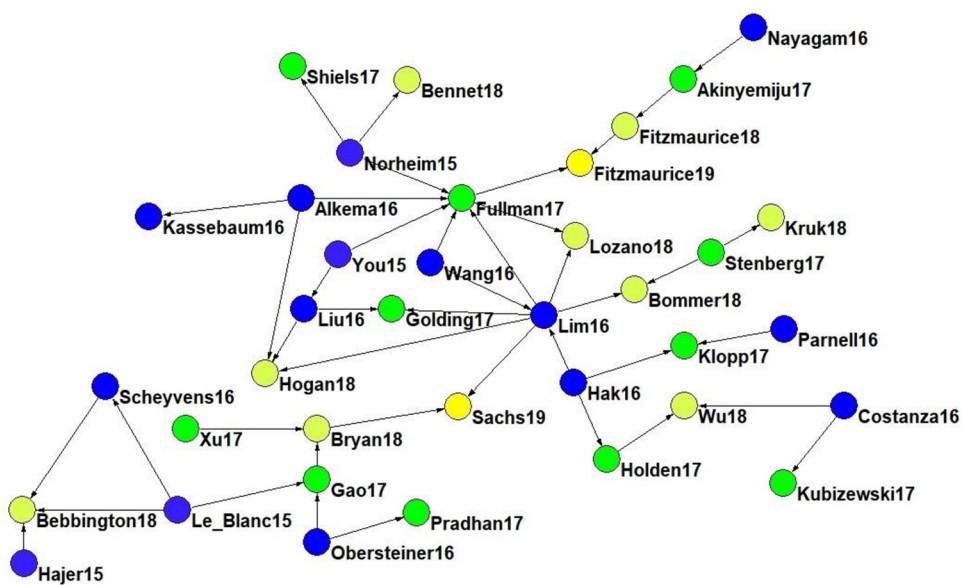

**Fig 5. Cross-citation graph.** Data source WoS. 2020.

in other articles in this set. The relationships between the articles in Fig 5 are complex and should be understood under a temporal sequence logic in the citation between two articles. However, some trends can be highlighted.

On the one hand, some contributions stand out for their centrality. Lim et al. [102] is connected with eight of the 37 articles (21.6%) on citing relationships, as is Fullman et al. [27], which relates to seven of the 37 articles (18.9%). Both authors researched health issues and are also coauthors of nine articles of the dataset under study. On the other hand, according to the SDG segmentation proposed, Hajer et al. [19] and Le Blanc [12] are recognized as seminal articles in social SDGs, since they contribute to the production of other subsequent articles in the set of 37. On the other hand, in health matters, seminal articles are Norheim et al. [103] and You et al. [104], two articles published in The Lancet whose citations also contribute to the production of the set introduced as Fig 5.

## 4. Discussion

The main purpose of this paper was to develop a critical and comprehensive scientometric analysis of the global academic literature on the SDGs from 2015 to 2020, conducted using the WoS database. The attained results have made it possible to comprehend and communicate to the scientific community the current state of the debate on the SDGs, thus offering insights for future lines of research.

To achieve the objectives, the present study analysed a broad spectrum of 5,281 articles published in 1,135 WoS journals. A first aspect that is striking is the great diversity of topics addressed in these studies, which reflects the multidimensionality of the SDGs. Despite this, more than half of the articles are concentrated in two JCR-WoS categories (Environmental Sciences and Green Sustainable Science Technology), a percentage that exceeds 80% if the categories Environmental Studies and Public Environmental Occupational Health are added. Thus, on the one hand, the size of the body of literature and the broad spectrum of topics more than covers the four perspectives of analysis that are relevant in research on the SDGs, according to Hajer et al. [19]: planetary boundaries, the safe and just operating space, the energetic society

and, last, green competition. However, on the other hand, results also highlight a strong focus on the environmental aspects of the SDGs, which undoubtedly concentrate the most contributions.

The Sustainable Development Goals constitute an area of research that has experienced exponential scientific growth, a tendency already suggested by previous studies [81, 105], thus complying with the fundamental principles of Price's law [76], which suggests the need for this exponential growth to manifest a continuous renewal of knowledge on the subject under study. The results of this study highlight a significant increase in the number of articles published in the last two years, given that six out of ten articles were published in 2019 or 2020. This tendency confirms how the SDGs continue to arouse great interest in the scientific community and that the debate on the interpretation of sustainable development is still open and very present in academia.

The variety of knowledge areas from which science can approach the SDGs demonstrates the different avenues that exist to address different research questions and their multidimensional nature, as anticipated by Pradhan et al. [17], a dispersion not far from the traditional fields of knowledge or the conventional dimensions of sustainability. Investigating the reasons for this dispersion in academic research on the SDGs may be a topic of great interest, as anticipated by Burford et al. [47] and Le Blanc [12], since understanding the phenomenon of development can only be achieved if the main challenges, both current and future, can be viewed holistically and comprehensively. Along these lines, Imaz and Eizagirre [106] state that the complexity of the study of the SDGs is undoubtedly marked by their aspiration for universality, by their broad scope encompassing the three basic pillars of sustainable development (economic development, environmental sustainability and social inclusion) and by their desire for integration, motivated by the complexity of the challenges and by the countless interlinkages and interdependencies.

This natural multidimensionality of the SDGs calls for strong cooperation and collaboration between researchers, universities, and countries. In this sense, the scientometric analysis provides good news, as more than a hundred prolific authors (defined as those authors who have published nine or more articles on this topic) have been identified, although these are reduced to eight in contemporary terms (2019 or 2020). This select group of eight authors who lead research and publishing on the SDGs (sometimes with dual or triple affiliations) produce knowledge for universities and research centres both in the global north and the global south: Canada, the U.S., the UK, Germany, Pakistan, Turkey, India, Benin, Russia and Cyprus. The protagonist role played by research institutes in countries in the north has already been acknowledged by previous studies [81, 105]. However, the emergence of top scholars producing academic knowledge from developing countries is a more recent tendency, which underscores the pertinence of this analysis.

A closer look at the academic and research curricula of these authors leads to the conclusion that the study of the SDGs does not constitute a final field of research at present. These researchers come from very heterogeneous disciplines, so their approach to the SDGs is also multidisciplinary. To illustrate it with an example, the most cited article by Professor Abhilash of Banaras Hindu University (the most published contemporary prolific author along with Christopher Murray of the University of Washington), with 363 WoS citations in February 2021 alone, is on the use and application of pesticides in India.

In more concrete terms, following Wu et al.'s [23] classification as a frame of reference, the eight most prolific contemporary authors approach the SDG research problem from two main domains, one of an environmental nature (Abhilash, Leal-Filho, Alola and Kalin) and the other related to health (Murray, Yaya, Bhutta, and Hay). The most common journals where these authors publish on environmental issues are the Journal of Cleaner Production, Higher

Education, Water and Science of the Total Environment. Health researchers, on the other hand, tend to publish mainly in the journals of the BMC group, The Lancet and Nature.

This wide diversity of academic fora can be clarified with the application of Bradford's laws, which identified a core of 18 journals that bring together the debates and academic discussions about the SDGs. It is worth noting that the 18 journals that form the core are distributed in 16 different thematic areas or WoS categories: Development Studies; Ecology; Economics; Education & Educational Research; Engineering, Environmental; Environmental Sciences; Environmental Studies; Green & Sustainable Science & Technology; Hospitality, Leisure, Sport & Tourism; International Relations; Medicine, General & Internal; Multidisciplinary Sciences; Public, Environmental & Occupational Health; Regional & Urban Planning; and Water Resources. On the one hand, this wide dispersion in terms of areas of knowledge suggests that research on the SDGs can be studied from different approaches and disciplines, which opens up a wide range of possibilities for researchers from different branches of scientific knowledge, as well as an opportunity for multidisciplinary collaborations. On the other hand, this heterogeneity might also hinder the communication and dissemination of learning from one field to another. The cross-citation analysis provided in Fig 5 suggests this possibility, as seminal works are related to thematic disciplines more than to the seminal contributions identified in Table 8.

In this sense, it is interesting to analyse the top-cited articles in the database, as they provide a clear picture of the field of knowledge. One-third of these contributions are provided by international institutions, such as the United Nations Development Program or the World Bank, which provide analyses of a normative nature. This prevalence reflects some weaknesses in the academic basis of the analysis of the SDGs as a whole from a scientific approach, an idea reinforced when the most cited papers are analysed. In fact, only six papers have reached more than 100 citations by contributions included in the database [4, 12, 24, 29, 101, 107]. Not only were these papers largely published before the approval of the SDGs themselves, but half of them are editorial material, inviting contributions but are not evidence-based research papers. Highlighting the nature of the most cited contributions does not diminish their value but does speak to the normative approach that underlies the analysis of the SDGs when addressed not individually but as an overall field of research.

Regarding topics and themes of interest, the scientometric analysis carried out in this research identified a strong concentration around a small number of terms, as represented in a heat map (Fig 2A and 2B). All these topics constitute a potential source of inspiration for future research on the subject.

Through an analysis of the main keywords, it can be seen that the studies focused on the traditional areas of health and climate change. However, these keywords also provide new elements for discussion, as they uncover some other areas of study that have been highlighted by the literature. First, the appearance of the term Management as one of the main keywords reveals the importance that researchers give to the role of business in achieving the SDGs, as already suggested by Scheyvens et al. [49] and Spangenber [22]. Second, the need to address new governance processes and to seek global solutions, as suggested by authors such as Sachs [4], underscore the keywords Governance, Policy and Framework, all aspects deemed crucial for the achievement of the SDGs and the 2030 Agenda [108]. Finally, other keywords such as Impact, Challenges or Systems are a clear example of the complexity and interdependencies that exist in research on the SDGs, considered an essential aspect by Griggs et al. [13] or Le Blanc [12]. The attained results highlight some of the connections between different domains of sustainable development by identifying categories and themes that are highly related in the groupings that emerge from the bibliographic coupling analysis.

In general terms, the holistic vision of development embodied by the SDGs has drawn the attention of very different disciplines, fields and areas of scientific knowledge. However, seven

major areas of research have emerged: environmental and public affairs, health, economics, health-burden of disease, economics-Kuznets curve, energy and soil-land. These areas are not far removed from the current paradigm of sustainable development, where poverty or inequality are problems that are not exclusive to developing countries [5, 6]. Thus, emerging issues that mainly affect first world countries, including urban planning, the impact of activities such as hospitality, sport or tourism, or education for development, are starting to stand out with increasing intensity, which continues to open new avenues for future research.

In short, the results of the scientometric analysis have provided a systematized overview of the research conducted in relation to the SDGs since the approval of the 2030 Agenda. Among other things, the critical analysis has identified the main trends with respect to the number of publications, the most relevant journals, the most prolific authors, institutions and countries, and the collaborative networks between authors and the research areas at the epicentre of the debate on the SDGs. As Olawumi and Chan [105] already acknowledged, the power research networks applied to the study of the SDGs offer valuable insights and in-depth understandings not only of key scholars and institutions but also about the state of research fields, emerging trends and salient topics.

Consequently, the results of this work contribute to the systematic analysis of scientific research on the SDGs, which can be of great interest for decision-making at the governmental level (e.g., which research to fund and which not to fund), at the corporate level and at the level of research centres, both public and private. Furthermore, the scientometric analysis carried out may provide clues for academics regarding future lines of research and topics of interest where the debate on the SDGs is currently situated.

## 5. Conclusions, limitations and future research lines

As could not be otherwise, all research in the field of social sciences has a series of limitations that must be clearly and transparently explained. The two most relevant in this study are the following.

First, although the study of the SDGs is a recent object of research, the rate of publication is growing exponentially, such that scientific knowledge is renewed practically in its entirety every two years. The only articles that escape this scientometric obsolescence are those with a high number of citations (h-index). This circumstance generates a temporal limitation in terms of the conclusions obtained in the present investigation, conclusions that should be revised periodically until the growth of publications stabilizes by adopting a logistic form, as recommended by Sun and Lin [109].

Second, the articles used as the basis for this research were restricted to those published in the JCR-WoS. This decision was made for two main reasons. On the one hand, the limitation was to eliminate potential distortions that could occur as a result of the constant growth of journals that are incorporated annually into other databases, such as ESCI-WoS (Emerging Sources Citation Index). On the other hand, it is impossible to compare impact indices if integrating other databases such as Scopus.

We are aware of these limitations, which for developing a more selective analysis imply assuming the cost of less coverage in exchange.

Regarding future lines of research, the analysis highlights how the study of the SDGs is failing to balance their economic, social and sustainability components, as it still maintains an overall focus on environmental studies.

This suggests the urgency of increasing studies on social SDGs, key topics on the 2030 Agenda including equity (SDGs 4, 5 and 10), social development (SDGs 11 and 16) and governance (SDG 17). These topics are part of the public discourse and currently a source of social pressure in many latitudes, but they are still research areas that are necessary to deepen.

Economic sustainability studies are more present, but highly concentrated, in health economics, as previously acknowledged by Meschede [81]. Academic research on the SDGs against poverty (SDG 1) and hunger (SDG 2) has not achieved such a prominent place as health. Even less so, the economics of technological development (SDGs 8 and 9), which are recognized as crucial for economic development.

Finally, the environmental SDGs do not achieve a balance among themselves either. Academic research has prioritized action for climate (SDG 13) and industrial and human consumption, mainly water (SDG 6) and energy (SDG 7). New research should be developed in the area of land (SDG 15), life under the sea (SDG 14) and sustainable production (SDG 12).

## Supporting information

**S1 Dataset.**
(XLSX)

## Author Contributions

**Conceptualization:** Antonio Sianes, Alejandro Vega-Muñoz, Pilar Tirado-Valencia, Antonio Ariza-Montes.

**Investigation:** Antonio Sianes, Alejandro Vega-Muñoz, Pilar Tirado-Valencia, Antonio Ariza-Montes.

**Methodology:** Alejandro Vega-Muñoz.

**Supervision:** Antonio Sianes, Alejandro Vega-Muñoz, Pilar Tirado-Valencia, Antonio Ariza-Montes.

**Validation:** Antonio Sianes, Alejandro Vega-Muñoz, Pilar Tirado-Valencia.

**Writing – original draft:** Antonio Sianes, Alejandro Vega-Muñoz, Pilar Tirado-Valencia, Antonio Ariza-Montes.

**Writing – review & editing:** Antonio Sianes, Alejandro Vega-Muñoz, Pilar Tirado-Valencia, Antonio Ariza-Montes.

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
