## [Decision Letter · Decision Letter 0]

28 Dec 2021

PONE-D-21-28963Impact of the Sustainable Development Goals on the academic research agenda. A scientometric analysisPLOS ONE

Dear Dr. Tirado-Valencia,

Thank you for submitting your manuscript to PLOS ONE. After careful consideration, we feel that it has merit but does not fully meet PLOS ONE’s publication criteria as it currently stands. Therefore, we invite you to submit a revised version of the manuscript that addresses the points raised during the review process.

We look forward to receiving your revised manuscript.

Kind regards,

Qaiser Abbas

Academic Editor

PLOS ONE

Journal Requirements:

3. Please remove your duplicate figures uploaded in the manuscript. These will be automatically included in the reviewers’ PDF.

Additional Editor Comments (if provided):

The language quality looks ordinary, therefore, you are also requested to recheck whole paper carefully and do needful language editing

Reviewers' comments:

Reviewer's Responses to Questions

**Comments to the Author**

1. Is the manuscript technically sound, and do the data support the conclusions?

Reviewer #1: Yes

Reviewer #2: Yes

2. Has the statistical analysis been performed appropriately and rigorously? 

Reviewer #1: Yes

Reviewer #2: Yes

3. Have the authors made all data underlying the findings in their manuscript fully available?

Reviewer #1: Yes

Reviewer #2: Yes

4. Is the manuscript presented in an intelligible fashion and written in standard English?

Reviewer #1: Yes

Reviewer #2: Yes

5. Review Comments to the Author

Reviewer #1: Really I want to give thanks for your great work and really it’s a great methodology and nice ideas for the analysis and looking forward for more ideas and researches relate to the same related topics

Reviewer #2: This paper offers a scientometric analysis of published articles on the overall theme of sustainable development goals (SDGs) during 2015-2020. To put it in perspective, it is observed that the authors record a total of 5,281 papers contributed by over 22,000 individuals, of which nearly 60% were penned during just two years, 2019-2020. This paper examines, among other, journal concentration, thematic coverage, academic nature of the contributions, and the depth of contribution. From all accounts, it appears to be an exhaustive analysis.

Comments and suggestions are included in the referee report attached herewith, which can be sent to the authors in entirety.

6. PLOS authors have the option to publish the peer review history of their article (what does this mean?). If published, this will include your full peer review and any attached files.

Reviewer #1: **Yes: **Haitham Medhat Aboulilah

Reviewer #2: **Yes: **Syed M. Ahsan

---

## [Author Response · Author response to Decision Letter 0]

20 Jan 2022

Response to Additional Editor Comments (if provided):

The language quality looks ordinary, therefore, you are also requested to recheck whole paper carefully and do needful language editing

WE CONFIRM THAT WE HAVE RESENT THE PAPER TO AJE, WHICH ALREADY REVISED THE ADEQUACY OF ENGLISH BEFORE THE FIRST SUBMISSION. WE ATTACH THE CERTIFICATE OF SUCH REVISION. 

WE HAVE ALSO RECHECKED OURSELVES THE FINAL VERSION, FOR EVEN MORE CAREFUL CONFIRMATION. WE HAVE NOT FOUND ANY MAJOR FLAWS, BUT OF COURSE WE ARE OPEN TO A NEW REVISION OF THE LANGUAGE IF NECESSARY.

Other comments:

WE HAVE ENSURED SUCH REQUIREMENTS.

2. In your Data Availability statement, you have not specified where the minimal data set underlying the results described in your manuscript can be found. PLOS defines a study's minimal data set as the underlying data used to reach the conclusions drawn in the manuscript and any additional data required to replicate the reported study findings in their entirety. Upon re-submitting your revised manuscript, please upload your study’s minimal underlying data set as either Supporting Information files.

WE HAVE ATTACHED THE DATASET FOR AVAILABILITY.

3. Please remove your duplicate figures uploaded in the manuscript. 

DUPLICATES HAVE BEEN REMOVED.

Response to Referees:

PLEASE FIND OUR RESPONSES IN THE ATTACHED DOCUMENT.

---

## [Editor Report · Decision Letter 1]

2 Mar 2022

Impact of the Sustainable Development Goals on the academic research agenda. A scientometric analysis

PONE-D-21-28963R1

Dear Dr. Tirado-Valencia,

We’re pleased to inform you that your manuscript has been judged scientifically suitable for publication and will be formally accepted for publication once it meets all outstanding technical requirements.

This acceptance is subject to you making some relatively minor revisions: in particular, I am asking you a further effort to make the paper more intelligible and fluid to read. I suggest to find am excellent professional proof reader to make it up to an academic journal standard, and to provide a better visualization of the network graphs - they are blurred and really hard to read.

Kind regards,

Stefano Ghinoi, Ph.D.

Academic Editor

PLOS ONE
---

## [Editor Report · Acceptance letter]

4 Mar 2022

PONE-D-21-28963R1 

Impact of the Sustainable Development Goals on the academic research agenda. A scientometric analysis 

Dear Dr. Tirado-Valencia:

I'm pleased to inform you that your manuscript has been deemed suitable for publication in PLOS ONE. Congratulations! Your manuscript is now with our production department. 

Kind regards, 

on behalf of

Dr. Stefano Ghinoi 

Academic Editor

PLOS ONE